# Water Environmental Capacity Calculation Based on Control of Contamination Zone for Water Environment Functional Zones in Jiangsu Section of Yangtze River, China

**Qiuxia Ma** [1] , **Yong Pang** [1,*] **and Ronghua Mu** [2]

1   Institute of Environment Planning and Evaluation, Hohai University, Nanjing 210098, China; maqiuxia@hhu.edu.cn

2   Nanjing Water Facilities Management Center, Nanjing 210098, China; 171305020060@hhu.edu.cn

*   Correspondence: ypang@hhu.edu.cn; Tel.: +86-135-8510-0649

**Abstract:** In recent years, due to unsustainable production methods and the demands of daily life, the water quality of the Yangtze River has deteriorated. In response to Yangtze River protection policy, and to protect and restore the ecological environment of the river, a two-dimensional model of the Jiangsu section was established to study the water environmental capacity (WEC) of 90 water environment functional zones. The WEC of the river in each city was calculated based on the results of the water environment functional zones. The results indicated that the total WECs of the study area for chemical oxygen demand (COD), ammonia nitrogen ($NH_3$-N), and total phosphorus (TP) were 251,198 t/year, 24,751 t/year, and 3251 t/year, respectively. Among the eight cities studied, Nanjing accounted for the largest proportion (25%) of pollutants discharged into the Yangtze River; Suzhou (11%) and Zhenjiang (12%) followed, and Wuxi contributed the least (0.4%). The results may help the government to control the discharge of pollutants by enterprises and sewage treatment plants, which would improve the water environment and effectively maintain the water ecological function. This research on the WEC of the Yangtze River may serve as a basis for pollution control and water quality management, and exemplifies WEC calculations of the world's largest rivers.

**Keywords:** water environmental capacity; water environment functional zone; contamination zone

## 1. Introduction

Jiangsu Province, a Chinese province currently undergoing rapid economic development, is faced with serious problems with regard to its water environment [1–3]. In particular, drinking water sources, nature reserves, and control sections on the Yangtze River are at risk due to pollution. In the 21st century, water security issues have become of utmost importance [4], and mathematical models have been used to assist in water quality management [5].

In order to further protect the water ecological environment, it is necessary to study the water environmental capacity (WEC), which refers to the maximum amount of pollutants that can be contained in a water body while maintaining normal function [6,7], based on hydraulic modeling that can calculate the flow direction, flow volume, and water quality transport [8]. Widely used models include the Streeter-Phelos (S-P) model [9–12], the MIKE model [13], the Environmental Fluid Dynamics Code (EFDC) model [14], the Water Quality Analysis Simulation Program (WASP) model [15–17], the Quality Simulation Along River Systems (QUASAR) model [18], the Soil and Water Assessment Tool (SWAT) model, the Modular Three-dimensional Finite-difference Ground-water Flow Model (MODFLOW) [19], and the Stream Water Quality Model (QUAL2K) [20,21].

Some scholars have established mathematical models to calculate the WEC of the study areas. For instance, the WEC of the urban lakeside area of Lake Taihu was estimated by the total standard method, and the water quality was calculated with a zero-dimensional

model [22]. Considering the synergetic influence of point and surface sources, a nonlinear optimization allocation model was used to calculate the WEC with the genetic algorithm based on controlled section water quality standards [23]. The subsection summation model was used to simulate the WEC of the river based on the demands and changes in the hydrological conditions [24]. An adequate characterization of the hydraulic state of a river is fundamental to the success of any water quality model, since it strongly affects various kinetic processes [25]. However, the above three WEC calculation methods do not fully consider the influence of topography on hydrodynamic simulations. As a professional modeling software, MIKE takes into account both topography and hydrological conditions to improve the hydrodynamic simulation accuracy and further improve the water quality simulation accuracy. The calculation of WEC is beneficial to resolve contradictions among administrative regions [26]. A MIKE 11 model was established to study the WEC of Qinhuai River, the main tributary of the Yangtze River [27]. Also, a WEC calculation framework based on MIKE 11 for the regional water environment functional zones was built in a plain river network area [28].

Some scholars have already studied the WEC of the Yangtze River. The WEC of the Chongqing section of the river was calculated using the coupled 1-D and 2-D models [29]. In addition, Chen et al. [30] calculated the allowable discharge loads in the Jiujiang section of the river under different hydrological conditions and in different water quality restricted zones. Li et al. [31] measured the WEC of the Nantong section of the river. However, there has been little research on the recent WEC of the whole Jiangsu section of the Yangtze River.

The Jiangsu section of Yangtze River is located at an estuary and has a wide river surface, so a 0-D or 1-D model cannot be easily applied. In contrast, the MIKE 21 model can generate a 2-D simulation and is able to simulate the water quantity and quality of sewage outlet-type problems. Jia et al. [32] simulated the increase in the concentration of pollutants using the MIKE 21 model in the Caofeidian sea area, which is significantly affected by the surrounding terrain. Therefore, a MIKE 21 model was built to calculate the pollution zone under different discharge amounts in each water environmental function zone in the Jiangsu section of the Yangtze River, considering the pollution sources, topography, hydrological conditions, and water quality objectives. When the pollution zone did not affect the drinking water sources, nature reserves, and control sections, the maximum discharge quantity was taken as the WEC. The Yangtze River is the third-longest river in the world and the longest river in Asia. Research on the WEC of the Yangtze River can serve as a basis for pollution control and water quality management of sewage outlet problems, and may play an exemplary role in WEC calculation of the world's largest rivers.

## 2. Materials and Methods

### 2.1. Study Area

The study area is the section of the Yangtze River, the longest river in China, that is located in central Jiangsu Province (Figure 1). The area extends from the junction of Anhui Province and Jiangsu Province to the estuary of the river. The area is south of Yangzhou, Taizhou, and Nantong, north of Zhenjiang, Changzhou, Wuxi, and Suzhou, and crosses Nanjing. Jiangsu Province has a subtropical monsoon climate, which is characterized by significant temperature fluctuations and distinct seasons. Due to the influence of the monsoon climate, the area has abundant precipitation, i.e., 724–1210 mm annually. However, there are obvious regional differences in precipitation, with more in the east than in the west, and more in the south than in the north.

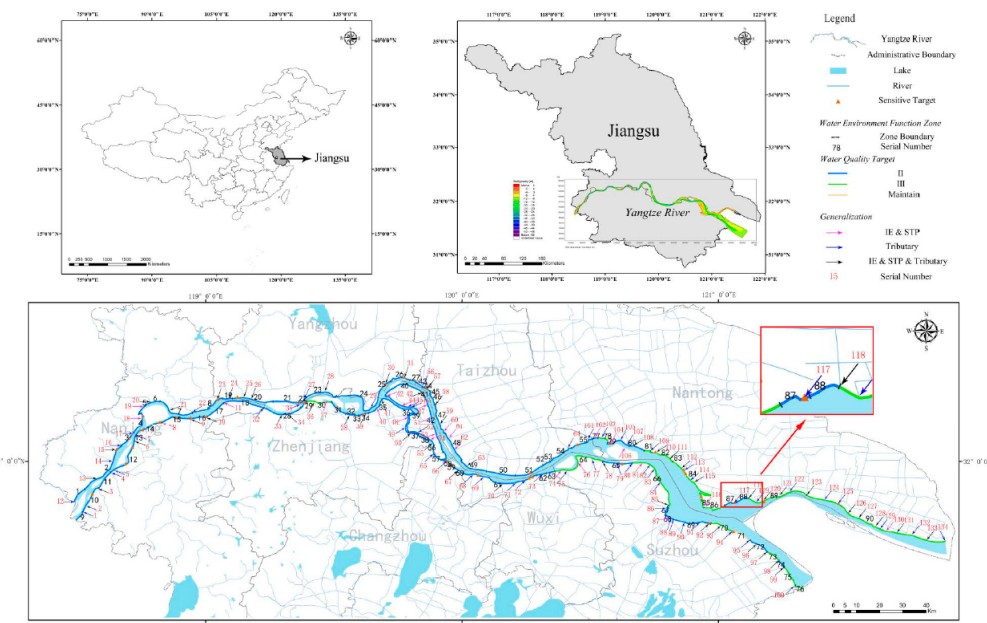

**Figure 1.** Study area and pollution source generalization map: industrial enterprises (IE); sewage treatment plants (STP).

There are 90 water environment functional zones in the Jiangsu section of the Yangtze River (as per the functional zoning of the surface water environment in Jiangsu Province, People's Government of Jiangsu Province, 2003), including 72 class II functional zones, 17 class III functional zones, and 1 functional zone that maintains current water quality. The chemical oxygen demand (COD), ammonia nitrogen (NH₃-N), and total phosphorus (TP) concentrations of class II zones are less than 15 mg/L, 0.5 mg/L and 0.1 mg/L, respectively, and those of class III zones are less than 20 mg/L, 1.0 mg/L and 0.5 mg/L, respectively. In addition, there are 40 sensitive targets (Figure 1).

*2.2. Data Collection*

Water quality data were collected from 26 sampling sites along the banks of the Yangtze River in 2015. The water samples were quickly transferred in a refrigerated state to the laboratory for analysis. All water quality indices were measured according to the China National Standard approach or the Environment Standard approach. Herein, the water quality indices consisted of COD, NH₃-N, and TP.

Data on the water level in the study area were obtained from the hydrological statistics of the Yangtze River Basin in the Annual Hydrological Report, People's Republic of China, in 2015 [33]. The underwater elevation of the model was extracted from the actual topographic computer-assisted design (CAD) graphic documents of the Yangtze River from Nanjing to Gaoqiao. Information on 199 industrial enterprises (IE) and 64 sewage treatment plants (STP) that discharge sewage directly into the Yangtze River was obtained from environmental statistics from 2017 for Jiangsu Province. Data on the water quality and quantity of the 111 main tributaries into the Yangtze River were provided by the Jiangsu Environmental Protection Bureau.

*2.3. Generalization of Sewage Outlet*

A statistical investigation revealed that the main sources of pollutants were industrial enterprises, sewage treatment plants, and the main tributaries along the Yangtze River. A total of 134 sewage outlets were generalized based on the environmental statistics of Jiangsu Province, including the geographical location and emission amount in 2017 (Figure 1). The equation for the discharge amount of each generalized outlet is as follows:

$$E_m^i = \sum_e E_m^{ie} + \sum_f E_m^{if} + \sum_g \left( C_m^{ig} \times Q_m^{ig} \right) \tag{1}$$

where $E_m^i$ is the total emission amount, $E_m^{ie}$ is the industrial enterprise emission amount, and $E_m^{if}$ is the sewage treatment plant emission amount of the $i$th generalized sewage outlet in the $m$th water functional zone (t/year). $C_m^{ig}$ is the water quality concentration in the tributary of the $i$th generalized sewage outlet in the $m$th water functional zone (mg/L). $Q_m^{ig}$ is the tributary flow of the $i$th generalized sewage outlet in the $m$th water functional zone (m$^3$/s).

*2.4. Water Environmental Capacity Calculation Methods*

2.4.1. Water Environmental Mathematical Model

A water environmental mathematical model of the Yangtze River was developed based on the MIKE 21 program by DHI. The governing equations of the 2-D hydrodynamics in a Cartesian coordinate system are continuity equations and momentum equations of integral Navier–Stokes mean equations of incompressible fluid in three dimensions along the direction of water depth. The continuity equation was expressed as:

$$\frac{\partial h}{\partial t} + \frac{\partial h\overline{u}}{\partial x} + \frac{\partial h\overline{v}}{\partial y} = hQ \tag{2}$$

The momentum equations were expressed as:

$$\frac{\partial h\overline{u}}{\partial t} + \frac{\partial h\overline{u}^2}{\partial x} + \frac{\partial h\overline{vu}}{\partial y} = f\overline{v}h - gh\frac{\partial \eta}{\partial t} - \frac{h}{\rho}\frac{\partial p_a}{\partial x} - \frac{gh^2}{2\rho}\frac{\partial \rho}{\partial x} + \\ \frac{\tau_{sx}}{\rho} - \frac{\tau_{bx}}{\rho} - \frac{1}{\rho}\left(\frac{\partial s_{xx}}{\partial x} + \frac{\partial s_{xy}}{\partial y}\right) + \frac{\partial}{\partial x}(hT_{xx}) + \frac{\partial}{\partial y}(hT_{xy}) + hu_sQ \tag{3}$$

$$\frac{\partial h\overline{v}}{\partial t} + \frac{\partial h\overline{uv}}{\partial x} + \frac{\partial h\overline{v}^2}{\partial y} = -f\overline{u}h - gh\frac{\partial \eta}{\partial y} - \frac{h}{\rho}\frac{\partial P_a}{\partial y} - \frac{gh^2}{2\rho}\frac{\partial \rho}{\partial y} + \\ \frac{\tau_{sy}}{\rho} - \frac{\tau_{by}}{\rho} - \frac{1}{\rho}\left(\frac{\partial s_{yx}}{\partial x} + \frac{\partial s_{yy}}{\partial y}\right) + \frac{\partial}{\partial x}(hT_{xy}) + \frac{\partial}{\partial y}(hT_{yy}) + hv_sQ \tag{4}$$

where $t$ is time (s), $x$, $y$ are Cartesian coordinates (m), $h$ is the total depth of the water (m), $\eta$ is water density (kg/m$^3$), $u$ and $v$ are the velocity components in the $x$ and $y$ directions, respectively (m/s). $f = 2\Omega\sin\varphi$ is Coriolis factor. $\tau_{sx}, \tau_{sy}$ are surface wind stress in the $x$ and $y$ directions, respectively (kg·m/s$^2$). $\tau_{bx}, \tau_{by}$ are base drag in the $x$ and $y$ directions, respectively (kg·m/s$^2$). $S_{xx}, S_{xy}, S_{yy}$ are radiation stress tensors. $P_a$ is atmospheric pressure (pa). $Q$ is point source emission flow (m$^3$/s). $g$ is the acceleration of gravity (m/s$^2$). $\rho$ is the density of water (kg/m$^3$). $u_s, v_s$ are the rates of external discharge into an environmental body of water (m/s).

Transverse stress $T_{ij}$ includes viscous resistance, turbulent friction resistance, and differential advection friction resistance, which can be calculated by the vorticity equation of the average vertical velocity:

$$T_{xx} = 2A\frac{\partial \overline{u}}{\partial x}, T_{xy} = A\left(\frac{\partial \overline{u}}{\partial y} + \frac{\partial \overline{v}}{\partial x}\right), T_{yy} = 2A\frac{\partial \overline{v}}{\partial x} \tag{5}$$

where $A$ is the area of contact (m$^2$).

The basic equation of convective diffusion of pollutants in two-dimensional, nonuniform flow can be expressed as:

$$\frac{\partial h\overline{C}}{\partial t} + \frac{\partial h\overline{u}\,\overline{C}}{\partial x} + \frac{\partial h\overline{v}\,\overline{C}}{\partial y} = h\left[\frac{\partial}{\partial x}\left(D_x\frac{\partial}{\partial x}\right) + \frac{\partial}{\partial y}\left(D_y\frac{\partial}{\partial y}\right)\right]\overline{C} + S + S_k \tag{6}$$

where $\overline{C}$ is the average pollutant concentration (mg/L), $\overline{u}$ and $\overline{v}$ are the velocity components in the $x$ and $y$ directions, respectively (m/s), and $D_x$ and $D_y$ are the diffusion coefficients in the $x$ and $y$ directions, respectively (m$^2$/s). $S$ is the source of pollution (g/m$^2$/s). $S_k$ is the dynamic conversion (g/m$^2$/s).

### 2.4.2. Designed Hydrological Conditions

According to the principle of selecting the most unfavorable hydrological conditions for the water environment, this calculation selected the water flow at Datong Hydrographic Station of the Yangtze River with a 90% hydrological guarantee rate after the construction of the Three Gorges Dam, comprehensively considering the influence of the Middle Route, the East Route of the South-to-North Water Diversion Project, and the water diversion project from Yangtze River to Huaihe River.

### 2.4.3. Model Parameter Sensitivity Analysis

A sensitivity analysis of the Manning number (N), COD degradation coefficient ($K_C$), ammonia nitrogen degradation coefficient ($K_N$), and total phosphorus degradation coefficient ($K_P$) in the model was carried out using the Morris screening method. The model was perturbed several times with a 5% step size, and the average value of the Morris coefficient was calculated using the following equation [34–37]:

$$S = \frac{\sum_{i=0}^{n-1} \frac{(Y_{i+1}-Y_i)/Y_0}{P_{i+1}-P_i}}{n-1}, \tag{7}$$

where $S$ is the sensitivity discriminant factor, $Y_{i+1}$ is the output value of the $i+1$-th run of the model, $Y_i$ is the output value of the $i$-th run of the model, is the initial value of the calculated result after parameter calibration, $P_{i+1}$ is the percentage of the parameter value after the $i+1$-th model operation relative to the calibration parameter, $P_i$ is the percentage of the parameter value after the $i$-th model operation relative to the calibration parameter, and $n$ is the total number of times the model ran.

### 2.4.4. Percentage Bias and the Nash–Sutcliffe Efficiency Coefficient

The calibration and validation of the model were carried out using the technique of graphic evaluation and error evaluation.

As a graphic evaluation technique, the time series diagram compared the measured value with the simulated value to judge the accuracy of the latter.

The model error was evaluated by calculating the percentage bias (PBIAS) and Nash–Sutcliffe efficiency coefficient (NSE) between the calculated value and the measured value. The PBIAS and NSE were calculated as follows [9,38–40]:

$$PBIAS = \frac{\sum_{i=1}^{n} (Y_{is} - Y_{io})}{\sum_{i=1}^{n} Y_{io}} \times 100\%, \tag{8}$$

$$NSE = 1 - \frac{\sum_{i=1}^{n} (Y_{io} - Y_{is})^2}{\sum_{i=1}^{n} (Y_{io} - \overline{Y_{io}})^2}, \tag{9}$$

where $PBIAS$ is the percentage bias (%), $NSE$ is the Nash-Sutcliffe efficiency coefficient, $Y_{is}$ is the simulated value of the model with time $i$ (the units of water flow is m$^3$/s, water quality is mg/L), $Y_{io}$ is the observed value of the model with time $i$ (the units of water flow is m$^3$/s, water quality is mg/L), and $\overline{Y_{io}}$ is average the observed value of the model with time $i$ (the units of water flow is m$^3$/s).

### 2.4.5. Water Environmental Capacity Calculation

Taking the emission amount of each generalized outlet as the background value, the contamination zone was calculated by importing the pollution source at the generalized outlet in the established mathematical model of the water environment. Then, the WEC of each water function was calculated as follows in Figure 2.

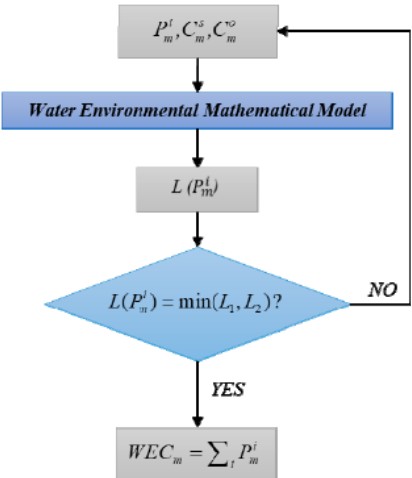

**Figure 2.** Calculation flow chart of the water environmental capacity (WEC).

Where $P_m^i$ is the pollution source of the *i*th generalized sewage outlet in the *m*th water functional zone (t/year), $C_m^s, C_m^o$ are the water quality target and current water quality, respectively, in the *m*th water functional zone (mg/L), $L(P_m^i)$ is the maximum longitudinal length of the contamination zone in the $P_m^i$ sewage outlet (m), $L_1$ is 10% of the length of the water functional zone (m), $L_2$ is the distance between the $P_m^i$ sewage outlet and the nearest sensitive target, which is a drinking water source, nature reserve, or control section (m), and $WEC_m$ is the *WEC* in the *m*th water functional zone (t/year).

## 3. Results

### 3.1. Generalization of Direct Discharge Pollution Sources

In 2017, 33,547 tons of COD, 1627 tons of NH$_3$-N, and 178 tons of TP were discharged directly into the Yangtze River water environment functional zones in eight cities along the river by industrial enterprises and sewage treatment plants (Figure 1). Nanjing directly discharged the most pollution into the Yangtze River; the amounts of COD, NH$_3$-N, and TP were 8924 t/year, 464 t/year, and 56 t/year, accounting for 26.6%, 28.5%, and 31.5% of the total, respectively (Table 1). Nantong discharged the second-highest amount of pollution, while Wuxi discharged no pollutants, directly.

**Table 1.** Emissions of pollutants from industrial enterprises and sewage treatment plants.

| City | COD (t/year) | NH$_3$-N (t/year) | TP (t/year) |
|---|---|---|---|
| Nanjing | 8924 | 464 | 56 |
| Yangzhou | 2666 | 47 | 17 |
| Zhenjiang | 3457 | 276 | 15 |
| Taizhou | 1610 | 55 | 19 |
| Changzhou | 6059 | 258 | 9 |
| Wuxi | 0 | 0 | 0 |
| Suzhou | 3646 | 203 | 19 |
| Nantong | 7185 | 323 | 43 |
| SUM | 33,547 | 1626 | 178 |

COD: chemical oxygen demand; NH$_3$-N: Ammonia nitrogen; TP: total phosphorus.

### 3.2. Model Parameter Sensitivity Analysis Results

According to a model parameter sensitivity analysis, N was an insensitive parameter, while K$_C$, K$_N$, and K$_P$ were high sensitivity parameters [34,35,37] (Table 2).

**Table 2.** Analysis of parameter sensitivity.

| Parameter | $|S|$ | Sensitivity Level | Grading Standard |
|---|---|---|---|
| N | 0.012–0.046 | insensitivity | $0 \leq |S| < 0.05$, insensitivity |
| $K_C$ | 2.714–4.059 | high sensitivity | $0.05 \leq |S| < 0.2$, medium sensitivity |
| $K_N$ | 2.304–3.346 | high sensitivity | $0.2 \leq |S| < 1$, sensitivity |
| $K_P$ | 0.937–1.133 | high sensitivity | $|S| \geq 1$, high sensitivity |

N: Manning number; $K_C$: chemical oxygen demand degradation coefficient; $K_N$: ammonia nitrogen degradation coefficient; $K_P$: total phosphorus degradation coefficient; S: sensitivity.

### 3.3. Model Calibration and Validation Results

#### 3.3.1. Model Calibration Results

The measured water level data of Nanjing Station (NJ) and Zhenjiang Station (ZJ) from 15 October to 22 October 2015, and the measured water quality data of the drinking water source of Zhangjiagang (ZJG) and Honggang (HG) from 17 October to 19 October 2015, were used to calibrate the model.

According to the model calibration results, N ranged from 29.3 to 50.6, and the degradation coefficients of COD, $NH_3$-N, and TP were 0.2 $day^{-1}$, 0.15 $day^{-1}$, and 0.06 $day^{-1}$, respectively.

The PBIAS and NSE of the simulated water level were calculated according to the measured water level data of NJ and ZJ. The PBIAS and NSE of NJ were 8.3% and 0.910 (Figure 3a). The PBIAS and NSE of ZJ were 12.4% and 0.905 (Figure 3b).

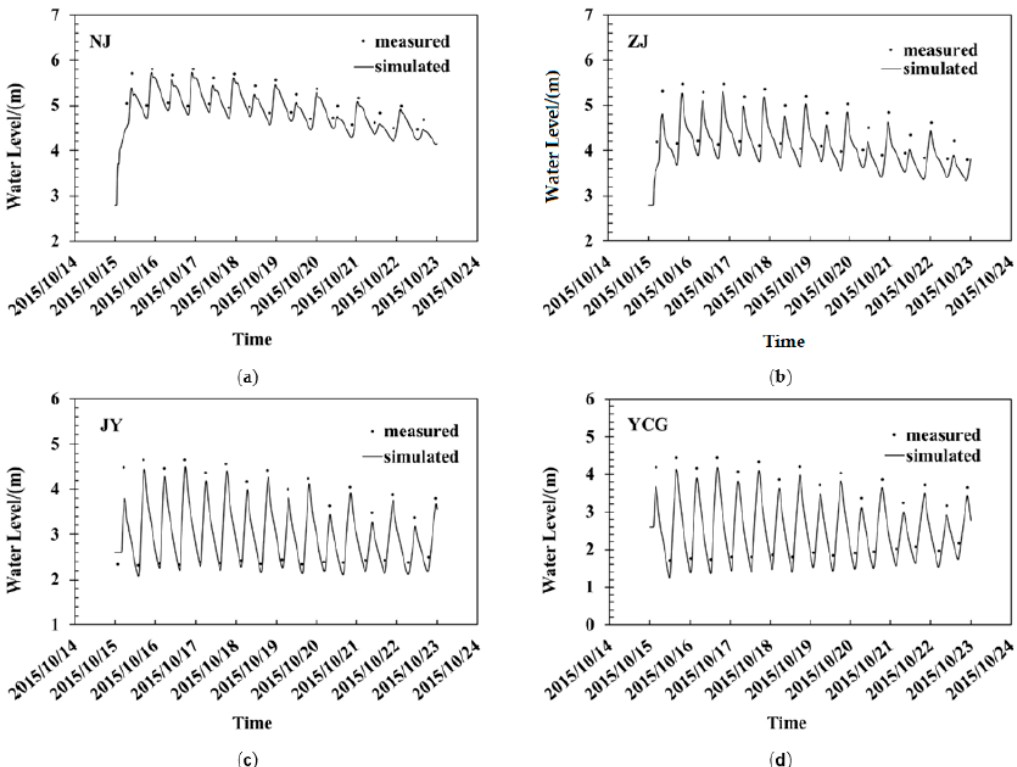

**Figure 3.** Calibration and validation for water level: (**a**) Nanjing (NJ); (**b**) Zhenjiang (ZJ); (**c**) Jiangyin (JY); (**d**) Yingchuangang (YCG).

The PBIAS of the simulated water quality was calculated according to the measured water quality data of the drinking water source, e.g., ZJG and HG. The average PBIAS of ZJG and HG are shown in Table 3. Using ZJG as an example, the time series diagrams are shown in Figure 4.

**Table 3.** The percentage bias (PBIAS) of the simulated water quality.

| Drinking Water Source | COD (%) | NH₃-N (%) | TP (%) |
|---|---|---|---|
| ZJG | 5.8 | 23.8 | 14.1 |
| HG | 5.1 | 29.7 | 17.9 |
| LG | 3.2 | 23.0 | 24.2 |
| LH | 1.3 | 3.8 | 16.4 |

COD: chemical oxygen demand; NH₃-N: Ammonia nitrogen; TP: total phosphorus. HG Honggang; LG Langgang; LH Liuhe; ZJG Zhangjiagang.

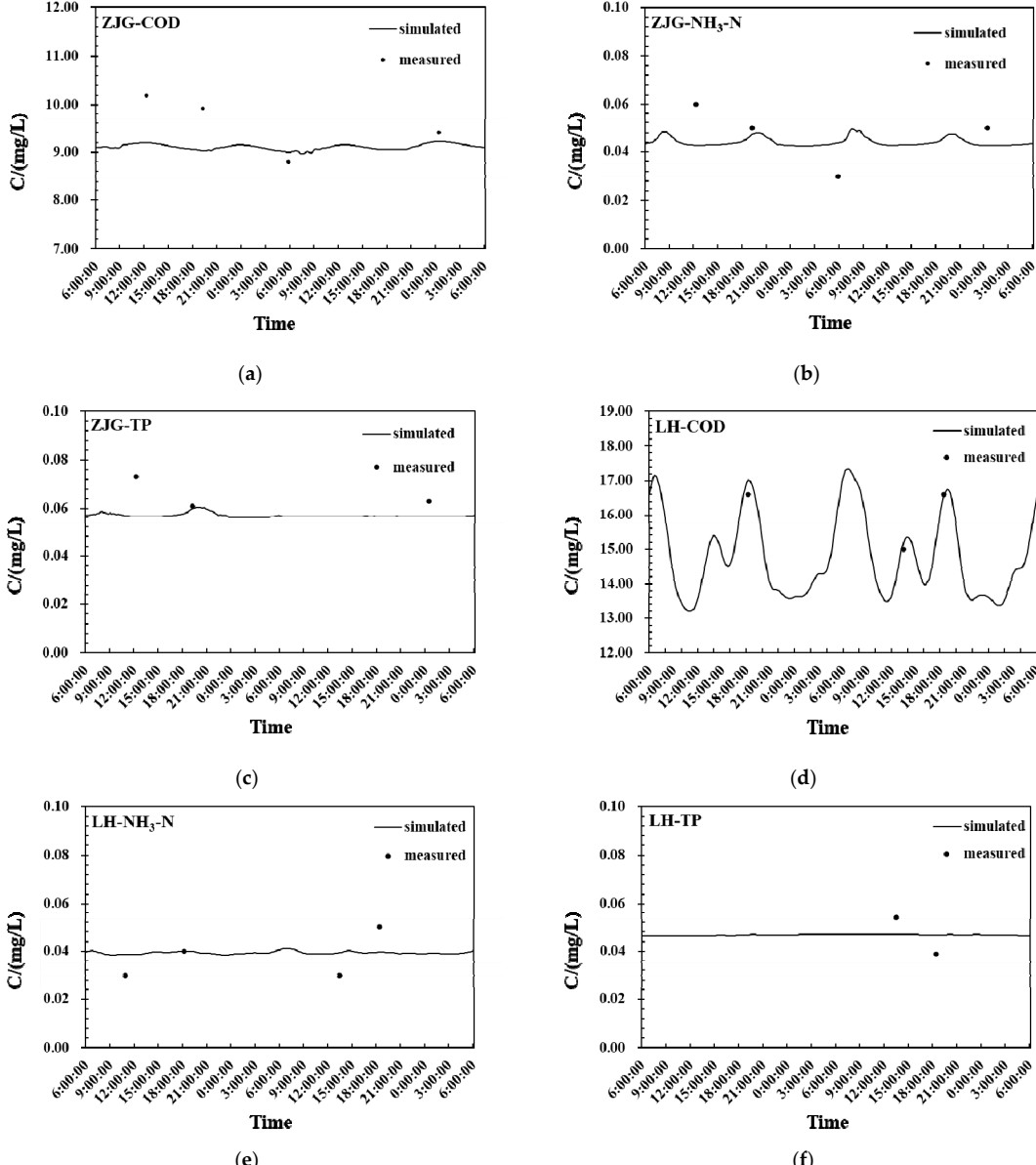

**Figure 4.** Calibration and validation for water quality: (**a**) Zhangjiagang (ZJG)-chemical oxygen demand (COD); (**b**) ZJG-ammonia nitrogen (NH₃-N); (**c**) ZJG-total phosphorus (TP); (**d**) Liuhe (LH)-COD; (**e**) LH-NH₃-N; (**f**) LH-TP.

### 3.3.2. Model Validation Results

The water level data of Jiangyin Station (JY) and Yingchuangang Station (YCG) from 15 October 2015 to 22 October 2015, and the measured water quality data of the drinking water source of Langgang (LG) and Liuhe (LH) from 17 October 2015 to 19 October 2015, were used to validate the model.

The PBIAS and NSE of the simulated water level were calculated according to the measured water level data of JY and YCG. The PBIAS and NSE of JY were 7.8% and 0.904 (Figure 3c). The PBIAS and NSE of YCG were 13.8% and 0.894 (Figure 3d).

The PBIAS of the simulated water quality was calculated according to the measured water quality data of the drinking water source, e.g., LG and LH. The average PBIAS of LG and LH are shown in Table 3. Using LH as an example, the time series diagrams are shown in Figure 4.

### 3.4. Calculation Results of Water Environmental Capacity

In accordance with the WEC calculation method, the capacity of 90 water environment functional zones was calculated using the established mathematical model (Figure 5). There were 251,198 tons, 24,751 tons, and 3251 tons of total capacities for COD, $NH_3$-N, and TP per year, respectively. Nanjing ranked first, with total capacities for COD, $NH_3$-N, and TP of 59,537 t/year, 8099 t/year, and 1008 t/year, accounting for 23.7%, 32.7%, and 31.0% of the total, respectively. Wuxi had the smallest capacities, with total capacities for COD, $NH_3$-N, and TP of 831 t/year, 121 t/year, and 15 t/year, accounting for 0.3%, 0.5%, and 0.5% of the total, respectively (Table 4).

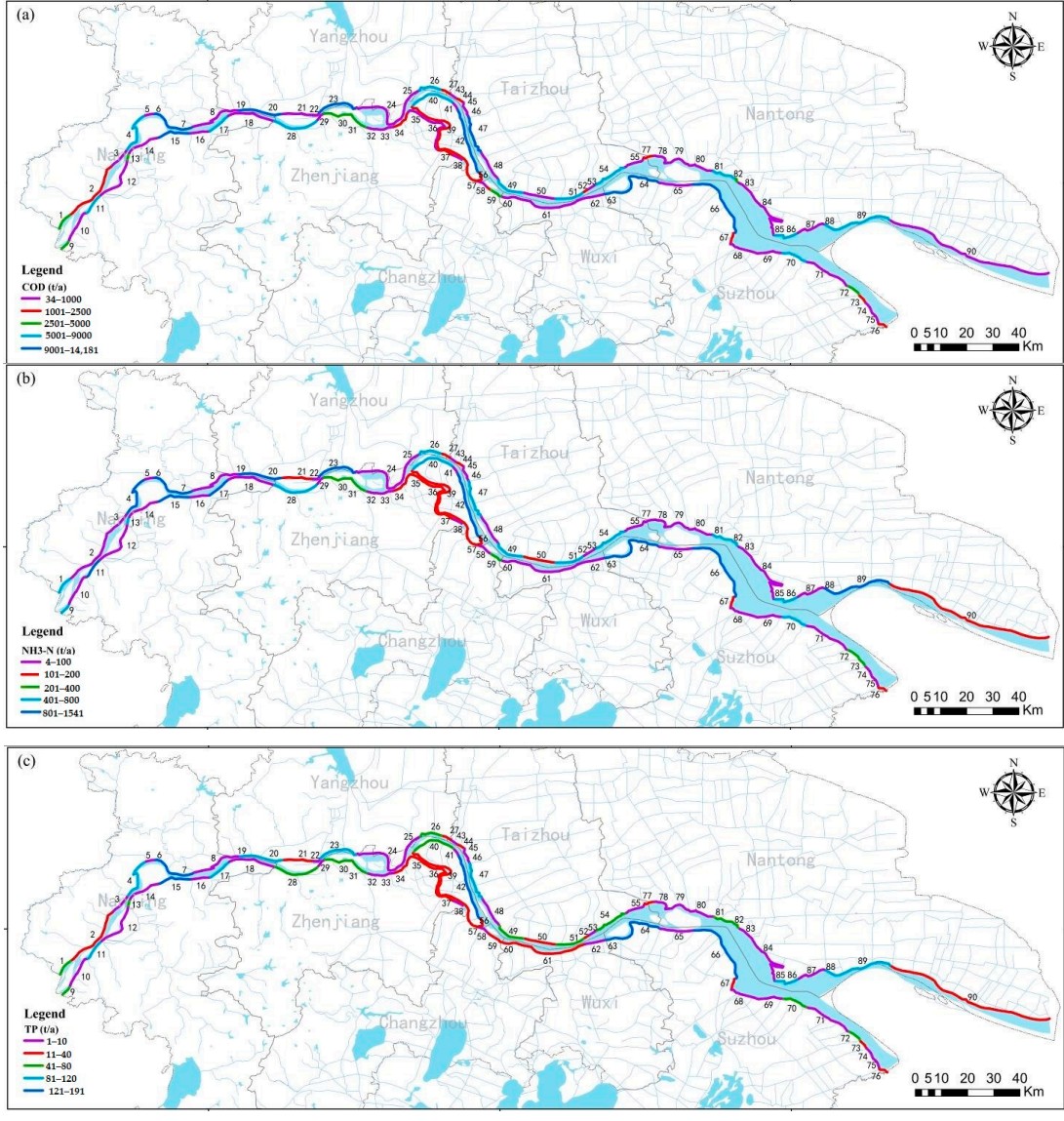

**Figure 5.** Water environmental capacity (WEC) of 90 water environmental function zones: (**a**) Chemical oxygen demand (COD); (**b**) Ammonia nitrogen (NH3-N); (**c**) Total phosphorus (TP).

**Table 4.** Water environmental capacity (WEC) of eight cities.

| City | COD (t/year) | NH$_3$-N (t/year) | TP (t/year) |
|---|---|---|---|
| Nanjing | 59,537 | 8099 | 1008 |
| Yangzhou | 26,830 | 2668 | 356 |
| Zhenjiang | 44,683 | 3344 | 480 |
| Taizhou | 37,351 | 2630 | 388 |
| Changzhou | 6689 | 538 | 76 |
| Wuxi | 831 | 121 | 15 |
| Suzhou | 42,384 | 4365 | 546 |
| Nantong | 32,893 | 2986 | 382 |
| SUM | 251,198 | 24,751 | 3251 |

COD: chemical oxygen demand; NH$_3$-N: Ammonia nitrogen; TP: total phosphorus.

## 4. Discussion

### 4.1. Sewage Outlet Generalization

The environmental statistics of Jiangsu Province showed that Nanjing had 32 industrial enterprises and 9 sewage treatment plants along the Yangtze River. In contrast, Wuxi had no industrial enterprises or sewage treatment plants along the river. Moreover, the industrial enterprises and sewage treatment plants in Nanjing produced a large amount of sewage discharge, which explained why Nanjing directly discharged the most pollutants while Wuxi discharged no pollutants (Table 1).

### 4.2. Model Performance Rating

Model performance can be judged on the grounds of general performance ratings [38,39] (Table 5). Based on the PBIAS of the water level in NJ and JY, the model performance can be rated "very good" and "good" for ZJ and YCG, respectively. Based on the NSE of the water level, the model performance can be rated "very good" for NJ, ZJ, JY, and YCG.

**Table 5.** General performance ratings for recommended statistics.

| Performance Rating | PBIAS (%) | | NSE |
|---|---|---|---|
| | Water Level | Water Quality | Water Level |
| Very Good | [−10, 10] | [−25, 25] | (0.75, 1] |
| Good | [−15, −10) ∪ (10, 15] | [−40, −25) ∪ (25, 40] | (0.60, 0.75] |
| Satisfactory | [−25, −15) ∪ (15, 25] | [−70, −40) ∪ (40, 70] | (0.40, 0.60] |
| Unsatisfactory | [−100, −25) ∪ (25, +∞) | [−100, −70) ∪ (70, +∞) | (−∞, 0.40] |

PBIAS: percentage bias; NSE: Nash–Sutcliffe efficiency coefficient.

In order to be consistent with the hydrodynamic calibration and validation time (15 October 2015 to 22 October 2015), the water quality data of the same period were selected in this study. The PBIAS for water quality in ZJG, LG, and LH received a "very good" rating, and the model performance can be rated "good" on the basis of the PBIAS in HG. Therefore, the established model was shown to be accurate and reliable, and can be used as the basis for WEC calculations.

### 4.3. Water Environmental Capacity

Based on the calculation of direct discharge pollutants earlier in the paper, Nanjing was found to be the city with the largest pollution discharge (Table 1). Nanjing was also the city with the largest WEC among the eight cities (Table 4), despite the fact that it had stricter water quality standards within the Yangtze River water environment functional area. According to the WEC calculation method, WEC was related to the water quality target, the maximum longitudinal length of the functional zone (L), sensitive targets, and the current sewage discharge. In order to prove that the calculation of WEC was reasonable, the length of the water functional areas was superimposed with the WEC and

the discharged contaminants of each city. The results showed that cities with longer water functional zones generally had a larger WEC (Figure 6).

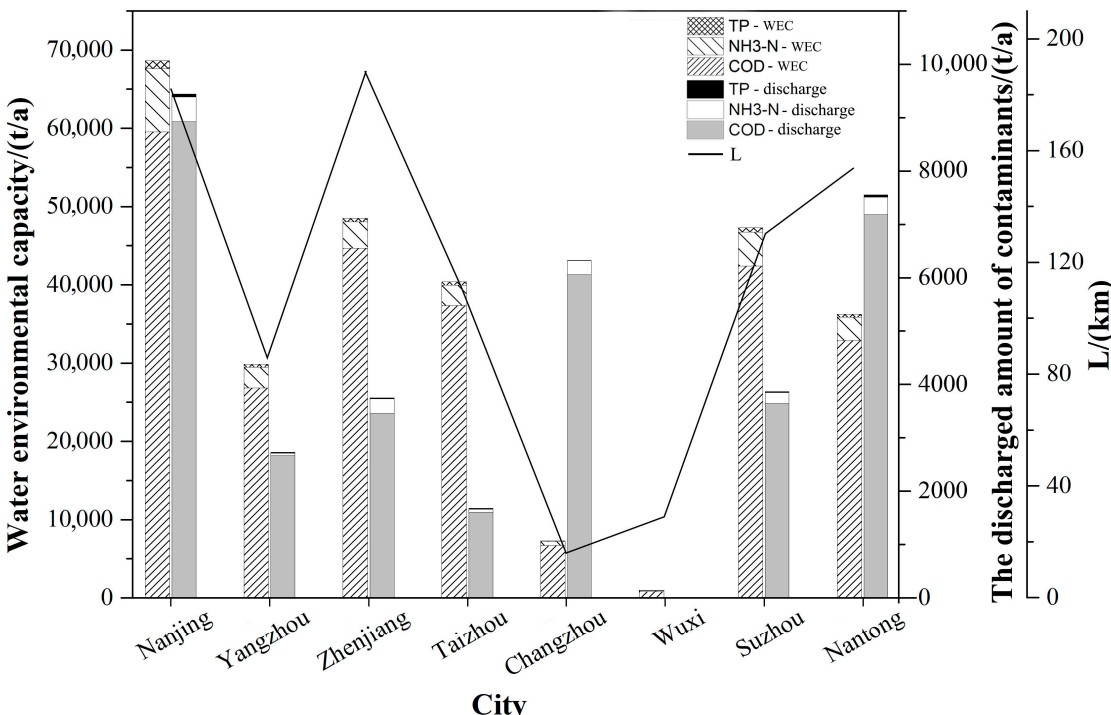

**Figure 6.** Relationship among the water environmental capacity (WEC) (left bars), the discharged amount of contaminants (right bars), and the maximum longitudinal length of the functional zone (L, solid line). COD Chemical oxygen demand; NH3-N Ammonia nitrogen; TP Total phosphorus.

Furthermore, under the condition of calculating the main pollution sources, the WEC calculation result was consistent with that of the Yangtze River Water Conservancy Commission, with a gap of less than 5%. The results of this calculation were scientific and reliable, and can provide technical support for pollution load reduction and water environment treatment and improvement in each water environment functional zone.

## 5. Conclusions

1. In order to respond to the national protection policy for the Yangtze River, a two-dimensional mathematical model of the water environment in the Jiangsu section was established to study the WEC of the Yangtze River. Based on the PBIAS of water level and quality, the model performance can be rated "good" or "very good," and the model can be well applied to the calculation of WEC.

2. The WEC was calculated on the condition that 10% of the length of the water environment functional zone was used as the sewage mixing zone, and the sensitive targets were not affected. There were 251,198 tons, 24,751 tons, and 3251 tons of total WEC for COD, $NH_3$-N, and TP per year, respectively. The calculation of the WEC in the 90 water environment functional zones of the river can provide a scientific basis for the protection and management of water resources, and lay a foundation for water resources planning.

3. The main tributaries, industrial enterprises, and sewage treatment plants that discharge sewage directly into the Yangtze River were considered in the process of sewage outlet generalization. Among them, the emissions of the main tributaries into the river changed with the water quality and quantity, and were not easily controlled. In contrast, the emissions from industrial enterprises and sewage treatment plants were less than the allowable emissions determined dialectically by national govern-

ment departments, and were generally fixed. Controlling the discharge of pollutants in the main tributaries will be the most important and difficult task in protecting the Yangtze River. Linking the allowable discharge of the main tributaries to the WEC of the Yangtze River has become an urgent challenge.

**Author Contributions:** Conceptualization, Q.M. and Y.P.; methodology, Q.M. and Y.P.; software, Q.M. and R.M.; resources, Y.P.; writing—original draft preparation, Q.M.; writing—review and editing, Q.M. and R.M.; visualization, Q.M.; project administration, Y.P.; funding acquisition, Y.P. All authors have read and agreed to the published version of the manuscript.

**Funding:** This research was funded by The National Science and Technology Major Project of China, grant number 2018ZX07208-005, and National Natural Science Foundation of China, grant number 51879070.

**Institutional Review Board Statement:** Not applicable.

**Informed Consent Statement:** Not applicable.

**Data Availability Statement:** Restrictions apply to the availability of these data. Data was obtained from Jiangsu Environmental Protection Bureau and are available from the authors with the permission of Jiangsu Environmental Protection Bureau.

**Acknowledgments:** We are very grateful to the anonymous reviewers and editors for their constructive comments.

**Conflicts of Interest:** The funders had no role in the design of the study; in the collection, analyses, or interpretation of data; in the writing of the manuscript, or in the decision to publish the results.

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
