# Peer review of "Water Environmental Capacity Calculation Based on Control of Contamination Zone for Water Environment Functional Zones in Jiangsu Section of Yangtze River, China"

_water, doi:10.3390/w13050587_

Round 1

Reviewer 1 Report

General comments

The manuscript Ma et al. “Water Environmental Capacity Calculation Based on Control of Contamination Zone for Water Environment Functional Zones in Jiangsu Section of Yangtze River, China” reports on the use of models for the calculation of Water Environmental Capacity (WEC) for the most downstream section of Yangtze River.

The study is well done and the manuscript is a well done report on the study, but not much more. The authors missed to indicate and demonstrate the novelty and the international relevance of their work. As long as this is not done, the manuscript has to be rejected because Water is an international journal while a Chinese journal would be the right place to reach the goal that the authors state at the end of the section Introduction: “This paper can serve as a basis for pollution control and water quality management and 80 may play an exemplary role in the calculation of the WEC of the Yangtze River in various provinces.” The fact that “there has been little research on the recent WEC of the whole Jiangsu section of the Yangtze River” is not reason enough for publication in an international journal although the Yangtze River is an internationally well-known and relevant water body. However, I am convinced that the authors are able to overcome the above discussed shortcomings of their manuscript. Therefore, I recommend “Rejection with invitation for re-submission after revision”. This evaluation is also the reason why I add some detailed comments on other, not that general shortcomings of the manuscript which require revision too.

Detailed comments

Lines 73-74: The statement applies only for the Jiangsu section of the Yangtze River but not for the entire river. Therefore, rewording is needed.

Lines 85-86: Also this sentence is not well prepared and not precise enough. I recommend revision as follows: “The study area is the section of the Yangtze River, the longest river in China that is located in central Jiangsu Province (Figure 1).”

Figure 1: All numbers provided in the figure are much too small and can hardly be read without a magnification glass. Enlargement is also required for the text of the legend. The meaning of “IE” and “STR” has to be explained in the heading of the figure.

Lines 97-101: An explanation of the meaning of the classes has to be added, at least a brief one in combination with a reference providing all details.

Eq. 3+4: The explanation of “τ” is missing and has to be added.

Eq. 3+4 and line 139: I wonder why the author do not use “ρ” for density as internationally common and as done for the water density at 4°C. I strongly recommend revision.

Line 141: In eq. 3 and 4, only “S” occurs without any index.

Eq. 5: The explanation of “A” is missing.

Eq. 6 and line 151: Why do the authors use “E” for the diffusion coefficient and not the internationally common “D”? I strongly recommend revision, in addition because “E” is used already as a variable in eq. 1.

Line 164: A reference for the Morris screening method should be provided.

Fig. 2: Why do the authors use “W” but not “WEC” in the flow scheme? This makes the reading needlessly complicated.

Line 207: I recommend replacing “for” by “of” at the beginning of the line.

Line 208: I recommend shifting “directly” to the end of the sentence.

Table 2: Adding a blank line after the table would be helpful.

Fig. 3: The numbers at the axes of the diagrams are much too small again. In the heading of the figure, the abbreviations of the names of the stations have to be added.

Fig. 4: Again, almost nothing can be read without magnification glass. Revision is needed. Furthermore, the figure shows that only very few measured data were available for calibration of the model. This issue is not discussed adequately in the section Discussion and requires the addition of a paragraph in subsection 4.2.

Fig. 5: The legends are much too small for good legibility and it would be helpful to use a bit thicker colored lines. Particularly green and yellow sections can hardly be seen and distinguished.

Fig. 6: The indices in the legend cannot be read because they are much too small. I strongly recommend replacing “amount of discharge” by “discharged amount of contaminants” in order to avoid any misunderstanding. “Discharge” usually means an amount of water. This replacement has to be done throughout the manuscript in the same way or analogously in order to use a uniform wording. In the heading of the figure, “(left bar)” should be added after “WEC”, “(right bar” after “amount of discharged contaminants” and “(solid line” after “L”. Finally, an explanation of the meaning of “L” has to be added to the heading.

Line 283: I recommend replacing “of” by “for”.

Lines 288-299: Much is rather a summarized repletion than a conclusion. Many details can be removed.

Reviewer 2 Report

A 2D model was used to simulate the water environment in the province of Jiangsu along the Yangtze river. Th water quality from 26 sampling stations were used for modeling. Overall, the manuscript is well presented and it fits the scope of the journal.

A few comments should be addressed for potential improvement.

  1. Many models are available to conduct the 2D simulation. Why the MIKE21 is used instead of WASP or QUAL?
  2. What are the impacts of precipitation on the simulation considering the length of the study area?
  3. In the data collection, some details should be given on the data quality and quantity.
  4. Model calibration and validation are missing, which are very important for for numerical simulations like this.
  5. The model results from Figure 3 did not show a good agreement with measured data, especially for figure 3d. Some uncertainty and statistical analysis could be helpful.

Reviewer 3 Report

Major comment: The introduction must be revised and extended to capture the significance of this work and how does it relate to former WEC studies. In particular, please expand WEC description, and its advantages, compared to other methods. You initially listed several hydrologic models (e.g. MODFLOW, SWAT, ...) but did not explain how they are related to WEC and your study. Please compare findings from former studies and highlight what it the novelty of your work. Minor comments: - Some models are missing full name (e.g. QUASAR, MODFLOW, QUAL2K) - Figure 1: numbers are very small and hard to read. Please use a bigger font size

Round 2

Reviewer 1 Report

The authors considered all my previous comments adequately. The changes probably done based on a further review also improved the manuscript.

A remaining need for revision is the formatting of page 3. This page is almost blank. The authors or the Editorial Office need to format teh manuscript in a way that such blank areas do not occur.

Therefore, I recommend acceptance after minor revision.

Reviewer 2 Report

Thanks for the response.

This reviewer is satisfactory with the changes made by the author.